# Model Problems on Oscillations of Mechanical and Biological Membranes

**Yury Kostikov [1] and Aleksandr Romanenkov [1,2,\*]** 

[1] Department 916 Mathematics, Moscow Aviation Institute; Volokolamskoye Shosse, 4, Moscow 125993, Russia
[2] Department 27 Mathematical Modeling of Heterogeneous Systems, Federal Research Center "Computer Science and Control" of the Russian Academy of Sciences, Vavilova, 44, Building 2, Moscow 119333, Russia
\* Correspondence: romanaleks@gmail.com; Tel.: +7-9262113029

**Abstract:** Various models of membrane oscillations emerging in the theory of elasticity of mechanical systems, biomechanics of the internal ear of vertebrata, and in the theory of electrical circuits are discussed in the article. The considered oscillations have different natures, but their mathematical models are described using similar initial boundary value problems for the second-order hyperbolic equation with the nontrivial boundary condition. The differential equations in these problems are the same. Thus, for example, the model of voltage distribution in the telegraph line emerges for the one-dimensional equation of oscillations. The model of oscillations of a circular homogeneous solid membrane, a membrane with a hole, and the model of gas oscillations in a sphere and spherical region emerge for the two-dimensional and three-dimensional operators, but take into account the radial symmetry of oscillations. The model problem on membrane oscillation can be considered as the problem on ear drum membrane oscillations. The unified approach to reducing the corresponding problems to the initial boundary value problem with zero boundary conditions is suggested. The technique of formulating the solution in the form of a Fourier series using eigenfunctions of the corresponding Sturm–Liouville problem is described.

**Keywords:** hyperbolic equation; modeling of oscillations with attenuation; boundary value problem; exact solutions



## 1. Introduction

The paper considers oscillatory processes with the presence of attenuation of various natures, the mathematical model of which is the initial boundary value problem for a linear equation of hyperbolic type. Attention is paid to the problems of voltage distribution in a telegraph line, the description of vibrations of a rectangular membrane, the description of vibrations of a membrane rigidly fixed at the edges in the multidimensional case, the description of radial gas vibrations in a spherically symmetric region, and the description of vibrations of a circular membrane, which, with some assumptions, can be considered as an approximation to the tympanic membrane of the ear of vertebrates. The obvious observation is that the sources of these tasks are various technical and biological processes.

The problems are conceptually different from the point of view of the nature of the processes under consideration. However, it is established that an important feature of these problems is the possibility of reducing them in a universal way to the same initial boundary value problem for a linear partial equation. Under certain assumptions, namely the fixed geometry of the domain, the assumption of the symmetry of the oscillations allows us to propose a unified approach and a method for constructing an accurate solution to emerging problems in the form of functional series.

It is worth noting that the multidimensional problems considered in the paper are successfully reduced to one-dimensional ones, that is, the dimension of the problem does not affect the process of constructing a solution, but it affects the type of eigenfunctions of the corresponding Sturm–Liouville problem. A feature in the formulation of the boundary

condition is the presence of an external non-stationary harmonic regime, namely, an external influence exerted on the oscillating system. This impact is not distributed within the area in which the solution of the corresponding problem is being sought, but is concentrated on the boundary of this area, that is, if the boundary conditions are not trivial. The method of special variable replacement was used in the work, which made it possible to reset the boundary conditions, and thus made it possible to apply the method of separation of Fourier variables.

For example, in works [1–5], mechanical models of vibrations of membranes and pendulum and elastic systems with friction are considered, and methods for analyzing the vibrational processes of these systems are proposed.

Let us point out that many authors pay attention to such problems. Thus, the problem of oscillation of a circular film membrane with electric current conductors distributed on it is discussed in [1]. The method of formulating the exact law of membrane oscillations was obtained with the help of the assumption of the solution linear dependence on radial variable and the Fourier method of separation of variables. The problem of controlling the string oscillation process without friction is considered in [2]. The mathematical models of dynamic processes in heterogeneous structures based on the hypotheses of complex rigidity and internal friction are considered in [3]. The exact solutions obtained in this work emerged when investigating ordinary differential equations with constant coefficients being the mathematical models of oscillations with friction. The method of obtaining the exact solution for the nonlinear model of oscillations with friction is proposed in [4]. The approximate-analytical method to calculate small free and forced oscillations of one-dimensional systems with dry friction is described in [5].

A mathematical model of the human auditory cochlea is given in [6]. This work shows that mathematical analysis of oscillations can be applied in the study of a biomechanical system. In [7], membrane oscillations are investigated in the presence of an external distributed disturbance. In [8], numerical methods are used to study vibrations and determine the qualitative effect of the appearance of an echo. It is worth noting that analytical methods are used in this work to accurately describe oscillatory processes, and the validity of their application can be found in [9,10]. Further analysis and discussion of the works will be carried out as necessary in the text.

In this work, a unified approach will be proposed, based on the method of separation of variables, which will allow for the investigation of model boundary value problems of hyperbolic type with a nonstationary boundary condition to present an exact solution in the form of a functional Fourier series for the corresponding system of basis functions.

A one-dimensional problem of voltage distribution in a telegraph line, a three-dimensional problem of radial oscillations of a gas in a spherical region, and a model problem of oscillations of a mammal's eardrum will be considered.

## 2. Materials and Methods

Let us consider the differential equation in partial derivatives:

$$\alpha^2(t)u_{tt} + 2\beta(t)u_t + \gamma^2(t)u = Lu, \tag{1}$$

$$u|_{x=0} = v(t), \quad u|_{x=l} = 0, \tag{2}$$

$$u|_{t=0} = u_t|_{t=0} = 0, \tag{3}$$

where $\alpha(t)$, $\beta(t)$, $\gamma(t)$ are set functions of time, the physical sense of which depends on the consideration of specific physical process, and $u = u(x, t)$—required function. The differential expression defined by the differential operator $L$ using spatial variables, which has the second order or higher, is in the first part of the Equation (1). In this article we will consider the case of one spatial variable and multidimensional problems, which can be reduced to a one-dimensional one, in some way or another.

Then, we will introduce the differential operator by the time variable:

$$D_t(\ldots) := \alpha^2(t)\frac{\partial^2}{\partial t^2}(\ldots) + 2\beta(t)\frac{\partial}{\partial t}(\ldots) + \gamma^2(t)(\ldots), \tag{4}$$

at the same time, the Equation (1) will be as follows:

$$D_t u = Lu, \tag{5}$$

where the near-boundary conditions (2) and initial conditions (3) remain unchanged.

Let us also point out that the Equation (5) can be considered not only as a model of oscillatory processes in mechanics but also in biology. It was found that the oscillatory process of an internal ear is described by a similar equation (see, for example, [6]). It is known that radial-symmetric oscillations are modeled by one-dimensional equations of hyperbolic type. Thus, the method of formulating the solution of the heterogeneous equation of (1) type at the harmonic type of heterogeneous component is proposed in [7]. "A very weak" solution is defined and the acoustic problem of shallow water, for which the qualitative effect of echo emergence is found using numerical methods, is considered in [8] for the equation similar to (1).

It is possible to substitute the required function so that the boundary conditions (2) will become trivial. For this we will introduce a new function $w(x, t)$:

$$u = w + v(t)\left(1 - \frac{x}{l}\right), \tag{6}$$

for which the Equation (5) and conditions (2)–(3) will be as follows:

$$D_t w + \left(1 - \frac{x}{l}\right)D_t v(t) = Lw, \tag{7}$$

$$w\big|_{x=0} = w\big|_{x=l} = 0, \tag{8}$$

$$w\big|_{t=0} = -v(0)\left(1 - \frac{x}{l}\right), \quad w_t\big|_{t=0} = -v'(0)\left(1 - \frac{x}{l}\right). \tag{9}$$

Then, we will substitute function $w$ for the summand with the first derivative by time variable to vanish in the Equation (7). Let

$$w = A(t)W, \quad v(t) = A(t)s(t), \tag{10}$$

where function $A(t)$ is defined from Cauchy problem for an ordinary differential equation:

$$\alpha^2(t)A'(t) + \beta(t)A(t) = 0, \quad A(0) = 1, \tag{11}$$

that is

$$A(t) = exp\left(-\int_0^t \frac{\beta(\tau)}{\alpha^2(\tau)}d\tau\right).$$

Then, according to (10) we have:

$$w = exp\left(-\int_0^t \frac{\beta(\tau)}{\alpha^2(\tau)}d\tau\right)W, \quad s(t) = exp\left(\int_0^t \frac{\beta(\tau)}{\alpha^2(\tau)}d\tau\right)v.$$

With such substitution we acquire the equation for the new function $W$:

$$\alpha^2(t)W_{tt} + Q(t)W + \left(1 - \frac{x}{l}\right)\left(\alpha^2(t)s''(t) + Q(t)s(t)\right) = LW, \tag{12}$$

where $Q(t)$ is defined by the Formula (13):

$$Q(t) = -\frac{\beta'(t)\alpha^2(t) - \beta(t)\left(\alpha^2(t)\right)'}{\alpha^2(t)} - \frac{\beta^2(t)}{\alpha^2(t)} + \gamma^2(t). \tag{13}$$

It should be noted that the boundary conditions (8) for the new function $W$ remain unchanged:

$$W|_{x=0} = W|_{x=l} = 0, \tag{14}$$

and the initial conditions will be as follows:

$$W|_{t=0} = -v(0)\left(1 - \frac{x}{l}\right), \quad W_t|_{t=0} = -\left(v'(0) + \frac{\beta(0)}{\alpha^2(0)}v(0)\right)\left(1 - \frac{x}{l}\right). \tag{15}$$

Let us point out that the solution of the Equation (1) is expressed through the solution of the Equation (12) by the formula:

$$u(x,t) = v(t)\left(1 - \frac{x}{l}\right) + exp\left(-\int_0^t \frac{\beta(\tau)}{\alpha^2(\tau)}d\tau\right)W(x,t). \tag{16}$$

Let us find the solution of the problem (12), (14), (15) with the help of the variables separation method. Let us consider the Equation (12), which does not contain a heterogeneous summand:

$$\alpha^2(t)W_{tt} + Q(t)W = LW, \tag{17}$$

where we find $W = T(t)X(x)$, then, after the insertion of this expression into the Equation (17), we have:

$$\alpha^2(t)\frac{T''}{T} + Q(t) = \frac{LX}{X}.$$

And the Sturm–Liouville problem emerges for the function $X = X(x)$:

$$LX = \lambda X, \quad X(0) = X(L) = 0. \tag{18}$$

As it is known, with the corresponding conditions for differential operator $L(\ldots)$, the problem (18) has the computation set of eigen values $\lambda_n$ and eigenfunctions $X_n(x)$. After that, we expand the function of one real variable $\left(1 - \frac{x}{l}\right)$ in a Fourier series by the system of eigenfunctions $X_n$ of the problem (18). Let

$$\left(1 - \frac{x}{l}\right) = \sum_n b_n X_n, \tag{19}$$

where $b_n$—Fourier coefficients found by the Formula (20):

$$b_n = \frac{\left(1 - \frac{x}{l}, X_n\right)}{\|X_n\|^2}. \tag{20}$$

Taking into account (19) and (20), the problem (12), (14), (15) will be as follows:

$$\sum_n \left(\alpha^2(t)T_n'' + Q(t)T_n + b_n\left(\alpha^2(t)s''(t) + Q(t)s(t)\right)\right)X_n = \sum_n \lambda_n T_n X_n,$$

$$W|_{t=0} = -v(0)\sum_n b_n X_n, \quad W_t|_{t=0} = -\left(v'(0) + \frac{\beta(0)}{\alpha^2(0)}v(0)\right)\sum_n b_n X_n.$$

Due to the linear dependence of own functions $X_n$, we use the family of Cauchy problems to correct $T_n(t)$ :

$$\alpha^2(t)T_n'' + (Q(t) - \lambda_n)T_n = -b_n\left(\alpha^2(t)s''(t) + Q(t)s(t)\right), \tag{21}$$

$$T_n|_{t=0} = -v(0)b_n, \quad (T_n)_t|_{t=0} = -\left(v'(0) + \frac{\beta(0)}{\alpha^2(0)}v(0)\right)b_n. \tag{22}$$

After solving them and with the help of the Formula (16) we obtain the exact solution of the initial problem (1)-(2)-(3) as a Fourier series:

$$u(x,t) = v(t)\left(1 - \frac{x}{l}\right) + exp\left(-\int_0^t \frac{\beta(\tau)}{\alpha^2(\tau)}d\tau\right)\sum_n T_n(t)X_n(x). \tag{23}$$

Let us briefly summarize the equations under consideration:

1. $\alpha^2(t)u_{tt} + 2\beta(t)u_t + \gamma^2(t)u = u_{xx}$—equations for voltage fluctuations in a limited telegraph line.
2. $\alpha^2(t)u_{tt} + 2\beta(t)u_t + \gamma^2(t)u = u_{xx} + u_{yy} + u_{zz}$, $(x, y, z): x^2 + y^2 + z^2 \leq R^2$—equation of damped gas oscillations in a spherical region.
3. $\alpha^2(t)u_{tt} + 2\beta(t)u_t + \gamma^2(t)u = u_{xx} + u_{yy}$, $(x, y): x^2 + y^2 \leq R^2$—vibration equation of a circular membrane.

The meaning of the parameters in the models under consideration in the general case is as follows: $\alpha^2(t)$—density, $\beta(t)$—coefficient that determines the resistance force proportional to speed, and $\gamma^2(t)$-viscosity coefficient.

## 3. Results

Following [9,10], let us consider the telegraph line with length $l$. With the distributed parameters $C$, $L$, $R$, $G$, where $C$—capacity per length unit, $L$—inductance per length unit, $R$—resistance per length unit, and $G$—conductivity per length unit (see [11,12]). Let the differential operator $L = \frac{\partial^2}{\partial x^2}$, $\alpha^2 = \sqrt{CL}$, $\beta = \frac{CR+LG}{2}$, $\gamma^2 = RG$. We accept that the right end of the line is grounded and the left one is connected to the power source, which supplies voltage according to the harmonic law:

$$v(t) = V\sin\omega t,$$

where $V$—voltage amplitude and $\omega$—frequency. We also assume that there is neither voltage nor current at the initial time moment. Let $u = u(x, t)$—voltage distribution in such telegraph line. It was established that this function is the solution of the homogeneous problem (1)-(2)-(3) (see, for example [13–15]).

With the entered parameters, it is simple to have that

$$Q(t) \equiv \gamma^2 - \frac{\beta^2}{\alpha^2}, \quad \lambda_n = -\left(\frac{\pi n}{l}\right)^2, \quad X_n(x) = \sin\frac{\pi n x}{l}, \quad b_n = \frac{2}{\pi n},$$

at $n \in \mathbb{N}$. The initial conditions (22) will be as follows:

$$T_n(0) = 0, \quad T_n'(0) = -2\frac{\omega V}{\pi n}. \tag{24}$$

In the Equation (21), the expression $\left(\gamma^2 - \frac{\beta^2}{\alpha^2} + \left(\frac{\pi n}{l}\right)^2\right)$ can be both positive and negative, but for all possible natural $n$ it will take the negative value only for a finite set of $n$ values.

Let $N_1 := \left\{ n \mid n \in \mathbb{N}, \gamma^2 - \frac{\beta^2}{\alpha^2} + \left( \frac{\pi n}{l} \right)^2 < 0 \right\}$. Then, let $-\sigma_n^2 = \gamma^2 - \frac{\beta^2}{\alpha^2} + \left( \frac{\pi n}{l} \right)^2$. Then, with $\forall n \in N_1$, we have

$$T_n = c_{1n} e^{\frac{\sigma_n t}{\alpha}} + c_{2n} e^{-\frac{\sigma_n t}{\alpha}} + \frac{\alpha^2 \omega^2 - \gamma^2}{\pi n (\alpha^2 \omega^2 + \sigma_n^2)} V \sin \omega t, \qquad (25)$$

and from the initial conditions we find that

$$c_{1n} = -\frac{\alpha}{2\sigma_n} \left( \frac{\omega V (\alpha^2 \omega^2 - \gamma^2)}{\pi n (\alpha^2 \omega^2 + \sigma_n^2)} + \frac{2\omega V}{\pi n} \right),$$

$$c_{2n} = \frac{\alpha}{2\sigma_n} \left( \frac{\omega V (\alpha^2 \omega^2 - \gamma^2)}{\pi n (\alpha^2 \omega^2 + \sigma_n^2)} + \frac{2\omega V}{\pi n} \right)$$

and

$$W = \sum_{n \in N_1} \left( \frac{\alpha}{2\sigma_n} \left( \frac{\omega V (\alpha^2 \omega^2 - \gamma^2)}{\pi n (\alpha^2 \omega^2 + \sigma_n^2)} + \frac{2\omega V}{\pi n} \right) \left( -e^{\frac{\sigma_n t}{\alpha}} + e^{-\frac{\sigma_n t}{\alpha}} \right) \right.$$
$$\left. + \frac{\alpha^2 \omega^2 - \gamma^2}{\pi n (\alpha^2 \omega^2 + \sigma_n^2)} V \sin \omega t \right) \sin \frac{\pi n x}{l}.$$

After that, relying on the Formulas (4) and (12), we have that

$$u(x, t) = \left( 1 - \frac{x}{l} \right) V \sin \omega t$$
$$+ \sum_{n \in N_1} \left( \frac{\alpha \omega V}{2\sigma_n} \left( \frac{\alpha^2 \omega^2 - \gamma^2}{\pi n (\alpha^2 \omega^2 + \sigma_n^2)} + \frac{2}{\pi n} \right) \left( -e^{\left( \frac{\sigma_n}{\alpha} - \frac{\beta}{\alpha^2} \right) t} + e^{-\left( \frac{\sigma_n t}{\alpha} + \frac{\beta}{\alpha^2} \right) t} \right) \right.$$
$$\left. + \frac{\alpha^2 \omega^2 - \gamma^2}{\pi n (\alpha^2 \omega^2 + \sigma_n^2)} V \sin \omega t \right) \sin \frac{\pi n x}{l}. \qquad (26)$$

Let now $N_2 := \left\{ n \mid n \in \mathbb{N}, \gamma^2 - \frac{\beta^2}{\alpha^2} + \left( \frac{\pi n}{l} \right)^2 > 0 \right\}$. Let us introduce the designation $\zeta_n^2 = \gamma^2 - \frac{\beta^2}{\alpha^2} + \left( \frac{\pi n}{l} \right)^2$. Then, with $\forall n \in N_2$ and $\omega^2 \neq \zeta_n^2$, we have

$$T_n = c_{1n} \sin \frac{\zeta_n}{\alpha} t + c_{2n} \cos \frac{\zeta_n}{\alpha} t + \frac{\alpha^2 \omega^2 - \gamma^2}{\pi n (\alpha^2 \omega^2 - \zeta_n^2)} V \sin \omega t, \qquad (27)$$

and from the initial conditions we find that

$$c_{1n} = -\frac{\alpha \omega V}{2\zeta_n} \left( \frac{\alpha^2 \omega^2 - \gamma^2}{\pi n (\alpha^2 \omega^2 - \zeta_n^2)} + \frac{2}{\pi n} \right), \quad c_{2n} = 0$$

and

$$W = \sum_{n \in N_2} \left( -\frac{\alpha \omega V}{2\zeta_n} \left( \frac{\alpha^2 \omega^2 - \gamma^2}{\pi n (\alpha^2 \omega^2 - \zeta_n^2)} + \frac{2}{\pi n} \right) \sin \frac{\zeta_n}{\alpha} t + \frac{\alpha^2 \omega^2 - \gamma^2}{\pi n (\alpha^2 \omega^2 - \zeta_n^2)} V \sin \omega t \right) \sin \frac{\pi n x}{l}.$$

Then, relying on the Formula (23), we have that

$$u(x, t) = \left( 1 - \frac{x}{l} \right) V \sin \omega t$$
$$+ \sum_{n \in N_2} \left( -\frac{\alpha \omega V}{2\zeta_n} \left( \frac{\alpha^2 \omega^2 - \gamma^2}{\pi n (\alpha^2 \omega^2 + \zeta_n^2)} + \frac{2}{\pi n} \right) \sin \frac{\zeta_n}{\alpha} t \right.$$
$$\left. + \frac{\alpha^2 \omega^2 - \gamma^2}{\pi n (\alpha^2 \omega^2 + \zeta_n^2)} V \sin \omega t \right) e^{-\frac{\beta}{\alpha^2} t} \sin \frac{\pi n x}{l}. \qquad (28)$$

For the resonance case $\alpha^2 \omega^2 = \zeta_n^2$, the solution defined by the Formula (27) will be replaced by

$$T_n = c_{1n} \sin \omega t + c_{2n} \cos \omega t + t (A \sin \omega t + B \cos \omega t). \qquad (29)$$

The constants from (29) can be also defined from the initial conditions and the formula for the exact solution in the form of a Fourier series, similar to (28), can be obtained.

Example 1. Model parameters $\alpha \equiv 1, \beta = 4,\ \gamma = 2$ (Figure 1):

$$u_{tt} + 8u_t + 4u = 25u_{xx}$$

$$u|_{x=0} = 2\sin 32\pi t,\ u|_{x=1} = 0$$

$$u|_{t=0} = u_t|_{t=0} = 0$$

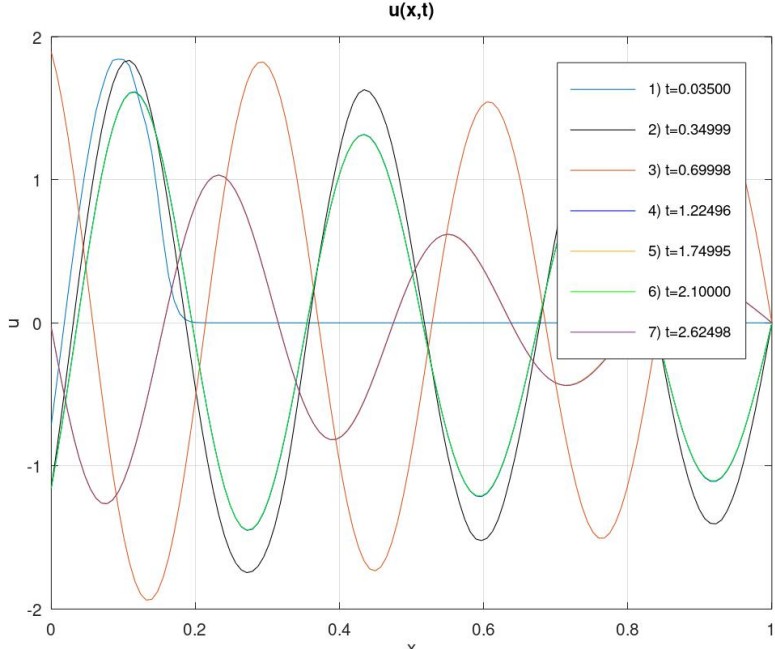

**Figure 1.** The graph of the solution at various points in time when $\alpha \equiv 1, \beta = 4,\ \gamma = 2$.

Example 2. Model parameters $\alpha \equiv 1, \beta = 16,\ \gamma = 2$ (Figure 2):

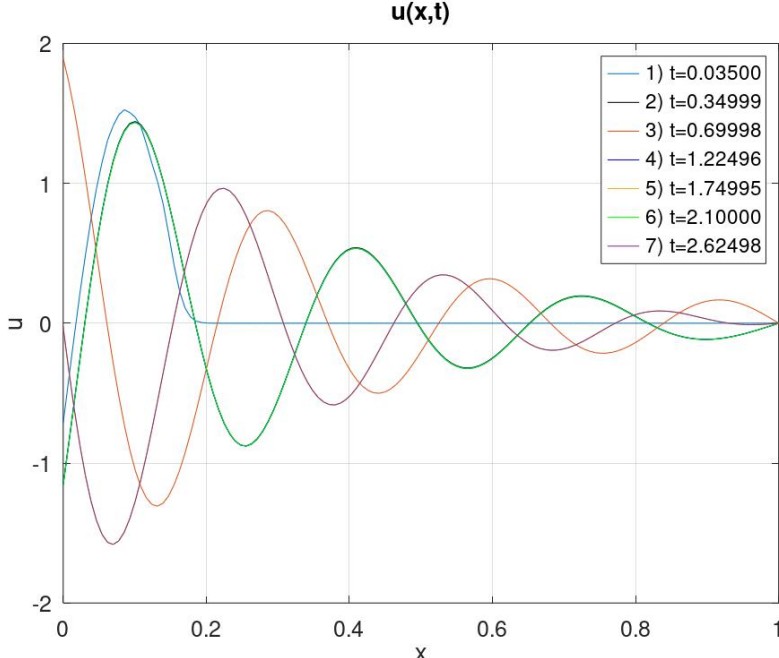

**Figure 2.** The graph of the solution at various points in time when $\alpha \equiv 1, \beta = 16,\ \gamma = 2$.

$$u_{tt} + 32u_t + 4u = 25u_{xx}$$

$$u|_{x=0} = 2\sin 32\pi t, \ u|_{x=1} = 0$$

$$u|_{t=0} = u_t|_{t=0} = 0$$

Example 3. Model parameters $\alpha \equiv 1, \beta = 0.8, \ \gamma = 0.2$ (Figure 3):

$$u_{tt} + 0.8u_t + 0.2u = 25u_{xx}$$

$$u|_{x=0} = 2\sin 32\pi t, \ u|_{x=1} = 0$$

$$u|_{t=0} = u_t|_{t=0} = 0$$

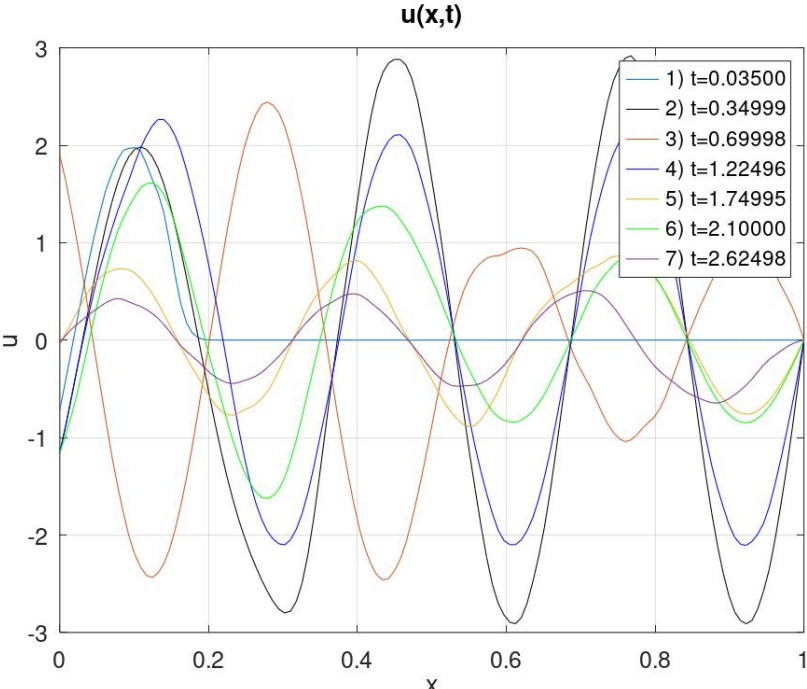

**u(x,t)**

**Figure 3.** The graph of the solution at various points in time when $\alpha \equiv 1, \beta = 0.4, \ \gamma = \sqrt{0.2}$.

Example 4. Model parameters $\alpha \equiv 1, \beta = 0.8, \ \gamma = 2$ (Figure 4):

$$u_{tt} + 0.8u_t + 2u = 25u_{xx}$$

$$u|_{x=0} = 2\sin 32\pi t, \ u|_{x=1} = 0$$

$$u|_{t=0} = u_t|_{t=0} = 0$$

Example 5. Model parameters $\alpha \equiv 1, \beta = 0, \ \gamma = 4$ (Figure 5):

$$u_{tt} + 4u = 25u_{xx}$$

$$u|_{x=0} = 2\sin 32\pi t, \ u|_{x=1} = 0$$

$$u|_{t=0} = u_t|_{t=0} = 0$$

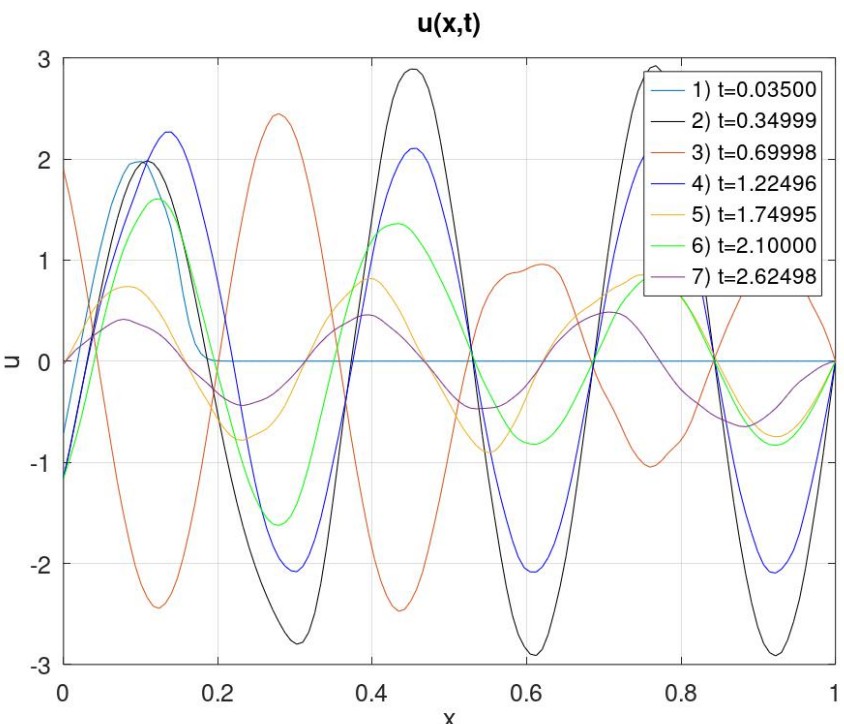

**Figure 4.** The graph of the solution at various points in time when $\alpha \equiv 1, \beta = 0.4, \gamma = \sqrt{2}$.

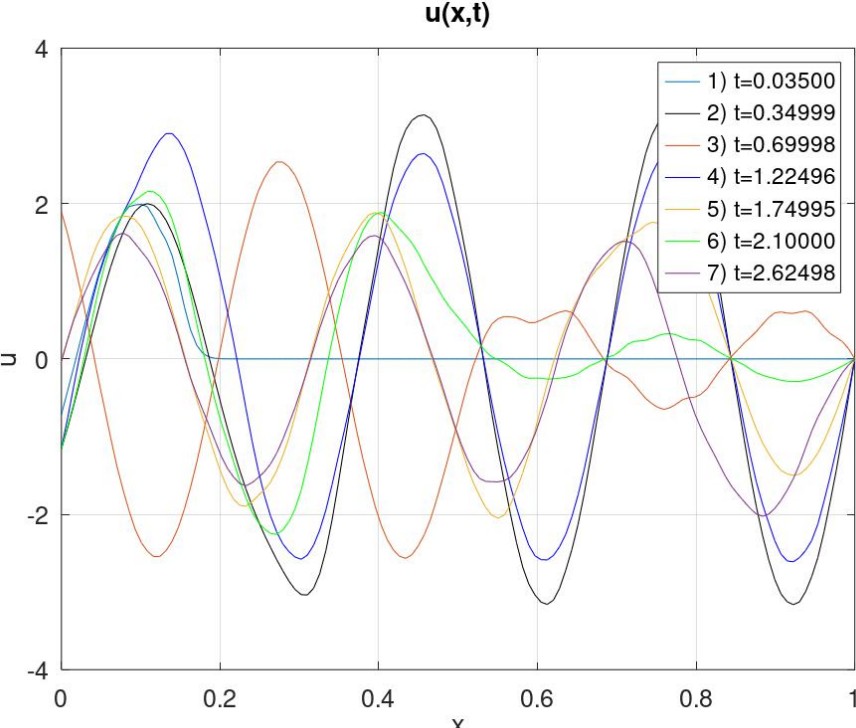

**Figure 5.** The graph of the solution at various points in time when $\alpha \equiv 1, \beta = 0, \gamma = 2$.

Example 6. Model parameters $\alpha \equiv 1, \beta = 32, \gamma = 0$ (Figure 6):

$$u_{tt} + 32u_t = 25u_{xx}$$

$$u|_{x=0} = 2\sin 32\pi t, \ u|_{x=1} = 0$$

$$u|_{t=0} = u_t|_{t=0} = 0$$

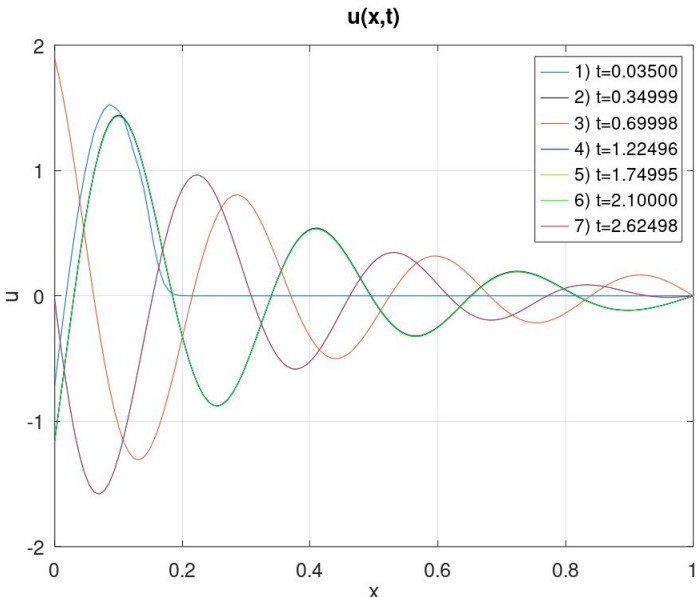

**Figure 6.** The graph of the solution at various points in time when $\alpha \equiv 1, \beta = 16, \gamma = 0$.

Example 7. Model parameters $\alpha \equiv 1, \beta = 0.2, \gamma = 0.2$ (Figure 7):

$$u_{tt} + 0.2u_t + 0.2u = 25u_{xx}$$

$$u|_{x=0} = 2\sin 32\pi t, \ u|_{x=1} = 0$$

$$u|_{t=0} = u_t|_{t=0} = 0$$

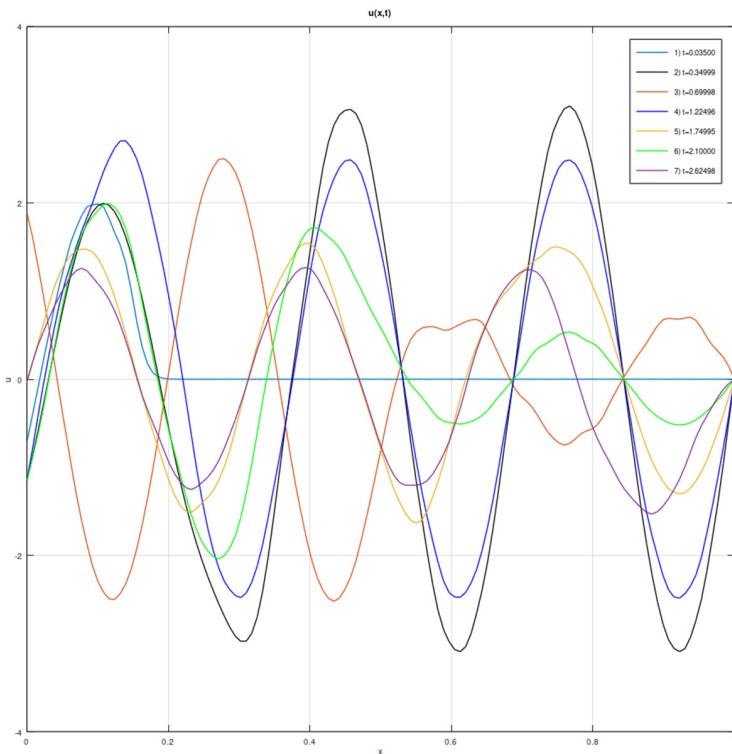

**Figure 7.** The graph of the solution at various points in time when $\alpha \equiv 1, \beta = 0.1, \gamma = \sqrt{0.2}$.

We should point out that a similar approach can be used for a two-dimensional case, when the region in which the oscillations are considered is square. The numerical-analytical solution of the problem of rectangular membrane oscillation was carried out in [16]. The oscillatory process was modeled; at the same time, it was demonstrated that the solution of the corresponding boundary value problem was defined by the double Fourier series:

$$u(x, y, t) = \sum_{k,\, j=1}^{\infty} U_{jk}(t) \sin \frac{\pi k x}{l_x} \sin \frac{\pi j y}{l_y},$$

where $l_x$, $l_y$—parameters of the rectangular region (length and width, respectively). At the same time, for the general multidimensional case and parallelepiped region in $\mathbb{R}^n$ with parameters $l_1$, $l_2$, $\ldots, l_n$, with differential operator $L(\ldots) = (\ldots)_{x_1 x_1} + (\ldots)_{x_2 x_2} + \cdots + (\ldots)_{x_n x_n}$, and conditions of rigid fixation at the boundary, the solution will be as follows:

$$u(x_1,\, x_2,\, \ldots,\, x_n, t) = \sum_{j_1,\, j_2,\, \ldots,\, j_n = 1}^{\infty} U_{j_1,\, j_2,\, \ldots,\, j_n}(t) \prod_{k=1}^{n} \sin \frac{\pi j_k x_k}{l_k}.$$

Eigen values for the considered problem are defined by the formula:

$$\lambda_{j_1,\, j_2,\, \ldots,\, j_n} = -\sum_{k=1}^{n} \frac{\pi^2 j_k^2}{l_k^2}.$$

### 3.1. Spherically Symmetric Cases in Three-Dimensional Space

Let us point out that the solution of the problem of small radial oscillations of gas with attenuation [17] with the availability of nonstationary disturbance at the boundary can be reduced to the considered problem (1)–(3). Let $u = u(x, y, z, t)$, $L = \Delta = \frac{\partial^2}{\partial x^2} + \frac{\partial^2}{\partial y^2} + \frac{\partial^2}{\partial z^2}$, and the region in which the oscillations occur be radially symmetrical. Thus, for example, it is possible to take the sphere with radius $R$ or spherical layer $\Omega = \{(x, y, z) | (x, y, z) \in \mathbb{R}^3,\ R_0^2 < x^2 + y^2 + z^2 < R_1^2\}$. It should be pointed out that the solution $u(x, y, z, t)$ will radially depend on spatial variables, i.e., the required function will actually depend on the distance to the origin of coordinates, namely, on the variable $r = \sqrt{x^2 + y^2 + z^2}$. Based on [18], we will write over the differential operator $L$ in radial coordinates:

$$u_x = u_r r_x = u_r \frac{x}{r},$$

$$u_{xx} = \left(u_r \frac{x}{r}\right)_x = u_{rx} \frac{x}{r} + u_r \left(\frac{x}{r}\right)_x = u_{rr} \frac{x^2}{r^2} + u_r \frac{r - \frac{x^2}{r}}{r^2} = u_{rr} \frac{x^2}{r^2} + u_r \left(\frac{1}{r} - \frac{x^2}{r^3}\right).$$

Due to the symmetry, we similarly obtain:

$$u_{yy} = u_{rr} \frac{y^2}{r^2} + u_r \left(\frac{1}{r} - \frac{y^2}{r^3}\right), \quad u_{zz} = u_{rr} \frac{z^2}{r^2} + u_r \left(\frac{1}{r} - \frac{z^2}{r^3}\right).$$

Then, $Lu = u_{xx} + u_{yy} + u_{zz} = u_{rr} \frac{x^2 + y^2 + z^2}{r^2} + u_r \left(\frac{3}{r} - \frac{x^2 + y^2 + z^2}{r^3}\right) = u_{rr} + \frac{2 u_r}{r}$. Now, the operator $L$ is finally defined by the Formula (30):

$$L := \frac{\partial^2}{\partial r^2} + \frac{2}{r} \frac{\partial}{\partial r}. \tag{30}$$

Let us formally set up the problem for a spherical region. It is necessary to define the solution of the equation

$$D_t u = \left(\frac{\partial^2}{\partial r^2} + \frac{2}{r} \frac{\partial}{\partial r}\right) u, \tag{31}$$

which satisfies the boundary conditions (32):

$$u|_{r=0} = 0, \quad u|_{r=R_0} = v(t), \tag{32}$$

and initial conditions (33):

$$u|_{t=0} = u_t|_{t=0} = 0. \tag{33}$$

Let us introduce the new function $Z = Z(r, t)$ by the Formula (34):

$$Z = ru, \tag{34}$$

which we use to obtain the equation for $Z$ with the help of (31). For this, it should be noted that $u = \frac{Z}{r}$ and $D_t u = \frac{1}{r} D_t Z$. Now, $Lu = L\left(\frac{Z}{r}\right) = \left(\frac{\partial^2}{\partial r^2} + \frac{2}{r}\frac{\partial}{\partial r}\right)\frac{Z}{r}$. Then, we calculate the derivatives in more detail:

$$\frac{\partial}{\partial r}\frac{Z}{r} = \frac{Z_r r - Z}{r^2} = \frac{Z_r}{r} - \frac{Z}{r^2},$$

$$\frac{\partial^2}{\partial r^2}\frac{Z}{r} = \frac{\partial}{\partial r}\left(\frac{\partial}{\partial r}\frac{Z}{r}\right) = \frac{\partial}{\partial r}\left(\frac{Z_r}{r} - \frac{Z}{r^2}\right) = \frac{Z_{rr} r - Z_r}{r^2} - \frac{Z_r r^2 - 2rZ}{r^4} = \frac{Z_{rr}}{r} - 2\frac{Z_r}{r^2} + 2\frac{Z}{r^3}.$$

Now, we obtain the recalculated expression for operator $L$:

$$L\left(\frac{Z}{r}\right) = \frac{Z_{rr}}{r} - 2\frac{Z_r}{r^2} + 2\frac{Z}{r^3} + \frac{2}{r}\left(\frac{Z_r}{r} - \frac{Z}{r^2}\right) = \frac{Z_{rr}}{r}.$$

We compile everything together and the Equation (31) is as follows:

$$D_t Z = Z_{rr}. \tag{35}$$

The boundary conditions (32) allow the identification of the boundary conditions for function $Z$ :

$$Z|_{r=0} = 0, \quad Z|_{r=R_0} = R_0 v(t), \tag{36}$$

and the initial conditions remain trivial:

$$Z|_{t=0} = Z_t|_{t=0} = 0. \tag{37}$$

The problem (35)–(37) coincides with the problem (1)–(3) practically completely, but it differs in the boundary condition. However, in this case, the replacement of the required function by the Formula (38):

$$Z = w + rv(t) \tag{38}$$

allows resetting of the boundary conditions to zero and, having repeated the solution of the problem (7)–(9), to obtain the precise expression for the function $Z(r, t)$, and then to write out the solution of the initial problem (31)–(33).

Only the boundary conditions will change for the spherical layer. Let the oscillation be absent on the external surface of the spherical layer and some mode, which depends only on the time variable, is set on the internal surface. In such case, we have the boundary conditions (39):

$$u|_{r=R_0} = v(t), \quad u|_{r=R_1} = 0. \tag{39}$$

In this case, the replacement of (34) remains in force, the Equation (35) does not change, and the boundary conditions (36) are as follows (40):

$$Z|_{r=R_0} = R_0 v(t), \quad Z|_{r=R_1} = 0, \tag{40}$$

and the initial conditions (37) remain unchanged. To reset the boundary conditions to zero, let us introduce the substitution (41):

$$Z = w + \left( \frac{r - R_1}{R_0 - R_1} \right) R_0 v(t), \tag{41}$$

that provides again the reduction to the previously solved problem (1)–(3).

### 3.2. Oscillations of Circular Membranes

Let us further consider the oscillations of a circular membrane. This model emerges when examining the oscillations of a human cochlea, as well as in technical acoustics [19]. The numerical experiment for modeling the oscillations of a circular membrane was conducted in [20]. A standard procedure of discretization of partial derivatives was performed in this work and the approximate solution was searched for using the numerical methods of algebra; however, the analytical investigation was not carried out for the considered problems. Surprisingly, the mathematical model of mammals' ear drum membrane [21] is the initial boundary value problem for the two-dimensional equation of oscillations [22]. The full-scale experiments connected with the investigation of drum membrane can be conducted, with certain accuracy, with the help of the formulas obtained in this work. The general provisions connected with the middle ear and drum membrane mechanics are described in the overview work [23], where it is demonstrated that the drum membrane can be interpreted as a mechanical membrane. In [24], the authors propose the system of two ordinary differential equations as a model of drum membrane oscillations, and the solution of this system is defined by an integral operator. The drum membrane model was examined and the probabilistic estimate of breakout risk as an explosion result was calculated in [25]. As before, the model of drum membrane oscillations set up using heterogeneous differential equations of the second order was considered. The transition to the model containing the equation in partial derivatives was performed in [26]. Here, the conclusion of the equation of drum membrane oscillations is made, the corresponding initially boundary value problem is set, and the analytical calculations with the application of special functions, namely Bessel functions, are given. The model with the equation in partial derivatives of hyperbolic type but with variable coefficients already is used in [27]. The proposed method allowed for the obtaining of numerical data compliant with the actual data, the source of which is the drum membrane physiology. The drum membrane dynamics with the fixed chain of ear bones is described with the help of the similar model in [28]. The articles [29–32] present the numerical investigation of the drum membrane oscillations. The numerical finite-element methods in modeling are given in [33,34]. The perspectives and possibilities of using model results are discussed, for example, in [35,36].

Let us dwell in more detail on some special situations that arise when considering specific values of the parameters of the model problem under consideration (1)-(2)-(3). In [22], vibrations of a viscoelastic annular plate are considered, which leads to the appearance of a fourth-order hyperbolic type equation. To solve this problem, the same method of expansion in terms of eigenfunctions is used. In the future, as a development of this work, a study of the influence of the term with the fourth order derivative will be carried out.

In article [24], the classical equation of forced harmonic vibrations without friction is considered as a simplified vibration model. So, if in our case for Equation (1) we take $Lu = F_{full}$ and assume that the solution u depends only on time, then for $\alpha^2(t) = m$, $\beta(t) = 0$, $\gamma^2(t) = k$, we exactly obtain Equation (8) from this work. Moreover, in the additional materials to this article, the Navier–Cauchy equations are explicitly obtained, which can be derived from Equation (1).

In [25], the equation of forced oscillations with friction is considered, in which the explicit form of the external influence is specified, namely: $F_{full} = AP_m \left( 1 - \frac{t}{t_0} \right) e^{-\frac{bt}{t_0}}$. Note that in this work the parameters in Equation (1) are defined as follows: $\alpha^2(t) = m$, $\beta(t) = \frac{c}{2}$, $\gamma^2(t) = k$, and a solution that depends only on time is considered.

Based on [26], we can put $\alpha^2(t) = \frac{\rho}{T}$, $\beta(t) = \gamma^2(t) = 0$, $L = \frac{\partial^2}{\partial r^2} + \frac{1}{r}\frac{\partial}{\partial r} + \frac{1}{r^2}\frac{\partial^2}{\partial \theta^2}$. These parameters correspond to the asymmetric vibrations of a round plate. Note that in the case of radial symmetric vibrations, the form of the basis functions will coincide with those obtained in this work.

In [27], the acoustics of the ear canal and eardrum are considered. The model of the processes under study is Equation (1) with $\alpha^2(t) = A(x)$, $\beta(t) = \gamma^2(t) = 0$, $L = \frac{c_0 \partial}{\partial x}\left(A(x)\frac{c_0 \partial}{\partial x}\right)$, with an additional inhomogeneous term of a special form.

In [28], a mathematical model is considered in which for Equation (1), $\alpha^2(t) = 1$, $\beta(t) = \gamma^2(t) = 0$, $L = c^2\left(\frac{\partial^2}{\partial r^2} + \frac{1}{r}\frac{\partial}{\partial r} + \frac{1}{r^2}\frac{\partial^2}{\partial \theta^2}\right)$ with nontrivial boundary conditions. The solution to the corresponding problem is obtained in the form of a Fourier functional series.

As it is known, in this case, the equation solution depends on two spatial variables, i.e., $u = u(x, y, t)$, we will investigate the radially symmetrical oscillations of the circular membrane of radius $R_0$. The assumption that the function $u = u(r, t)$, where $r = \sqrt{x^2 + y^2}$, will be used. Operator $L$ is recalculated in exactly the same way as before. And now we have that the differential operator $L$ is defined by the Formula (42):

$$L := \frac{\partial^2}{\partial r^2} + \frac{1}{r}\frac{\partial}{\partial r}, \tag{42}$$

and boundary conditions (2) are as follows:

$$u|_{r=R_0} = v(t), \quad |u|\big|_{r \leq R_0} < \infty. \tag{43}$$

The initial conditions (3) remain the same. The boundary conditions (43) are zeroed by the standard replacement of the variable that was previously used in this work. It is known that the operator's eigenfunctions (42) are zero-order Bessel functions of the first kind [17,18]:

$$X_n = J_0\left(\frac{\kappa_n r}{R_0}\right), \tag{44}$$

where $\kappa_n$—equation root

$$J_0(\kappa_n) = 0. \tag{45}$$

A Fourier series, which gives the solution of the corresponding boundary value problem, is defined by the formulas similar to (26) and (28). The difference is that Bessel functions will be the basic functions and sets $N_1$ and $N_2$ will be defined by the roots of the Equation (45). The solution kind is demonstrated by the formula (46):

$$u = rv(t) + \sum_{n \in M_1} T_{1n}(t) e^{-\frac{\beta}{\alpha^2}t} J_0\left(\frac{\kappa_n r}{R_0}\right) + \sum_{n \in M_2} T_{2n}(t) e^{-\frac{\beta}{\alpha^2}t} J_0\left(\frac{\kappa_n r}{R_0}\right), \tag{46}$$

where $M_1 := \left\{n \,|\, n \in \mathbb{N}, \gamma^2 - \frac{\beta^2}{\alpha^2} + \left(\frac{\kappa_n}{R_0}\right)^2 < 0\right\}$, $M_2 := \left\{n \,|\, n \in \mathbb{N}, \gamma^2 - \frac{\beta^2}{\alpha^2} + \left(\frac{\kappa_n}{R_0}\right)^2 > 0\right\}$, $T_{1n}(t)$—the function, which does not contain periodic summands relative to $t$, and $T_{2n}(t)$—the function, which contains periodic summands relative to $t$. It should be pointed out that set $M_1$ is finite or empty, set $M_2$ is countable.

Let us now consider the spherical membrane oscillations. In this case, $R_1$—membrane radius and $R_0$—radius of the hole in the membrane. Let us consider that the membrane is fixed by its edge, and some mode, which depends only on the time variable, is set on the internal boundary, i.e., the boundary conditions are fulfilled

$$u|_{r=R_0} = v(t), \quad u|_{r=R_1} = 0. \tag{47}$$

It should be pointed out that, in this case, the substitution of the variable (6) is not good and the following formula should be used:

$$u = w + v(t)\left(\frac{r - R_1}{R_0 - R_1}\right). \tag{48}$$

The following boundary conditions will be fulfilled for the function $w = w(r, t)$:

$$w|_{r=R_0} = w|_{r=R_1} = 0, \tag{49}$$

as well as the initial conditions (50):

$$w|_{t=0} = -v(0)\left(\frac{r - R_1}{R_0 - R_1}\right), \quad w_t|_{t=0} = -v'(0)\left(\frac{r - R_1}{R_0 - R_1}\right). \tag{50}$$

It should be noted that when substituting (33), the heterogeneous summand will change in the Equation (7). For the considered case we have:

$$D_t w + \left(\frac{r - R_1}{R_0 - R_1}\right) D_t v(t) - \frac{1}{R_0 - R_1} v(t) = Lw. \tag{51}$$

For this case, zero-order Bessel functions of the first and second kind are the eigenfunctions of operator $L$. At the same time, the eigenfunctions are defined by the Formula (52) for boundary conditions (49):

$$X_n = N_0(\kappa_n R_0) J_0(\kappa_n r) - J_0(\kappa_n R_0) N_0(\kappa_n r), \tag{52}$$

where $\kappa_n$—roots of the characteristic equation:

$$J_0(\kappa R_0) N_0(\kappa R_1) - J_0(\kappa R_1) N_0(\kappa R_0) = 0. \tag{53}$$

The desired solution of the problem on the circular membrane oscillation is defined by a Fourier series using the system of eigenfunctions (52). The Fourier series kind is defined by the formula similar to (46).

## 4. Conclusions

The article considers the mathematical model of oscillations with friction and set mode at the boundary, which depends only on time. The initial boundary value problem for the second-order hyperbolic equation with the nontrivial boundary condition describes the law of oscillations. The second-order differential operator considered in this work is either originally one-dimensional by time variables or, relying on the oscillatory process symmetry, can be reduced to one-dimensional. With the help of successful substitution and the method of separation of variables, we managed to reduce this problem solution to the known problem of one-dimensional bounded string oscillation and suggested the method of solution formulation as a Fourier series. At the same time, the obtained Fourier series, depending on the type of differential operator by time variables, set the voltage distribution for the telegraph line, law of gas oscillations in a sphere or spherical region, and the law of oscillation of a circular membrane without and with a hole. The work results can be applied in practical calculations of telegraph lines, modeling of membrane oscillations in acoustics, biology, and medicine.

**Author Contributions:** Conceptualization, Y.K. and A.R.; methodology, A.R.; software, A.R.; validation, Y.K.; formal analysis, Y.K.; investigation, A.R.; resources, Y.K.; data curation, Y.K.; writing—original draft preparation, A.R.; writing—review and editing, A.R.; visualization, A.R.; supervision, A.R.; project administration, Y.K. All authors have read and agreed to the published version of the manuscript.

**Funding:** This research received no external funding.

**Data Availability Statement:** Data are contained within the article.

**Conflicts of Interest:** The authors declare no conflict of interest. The funders had no role in the design of the study; in the collection, analyses, or interpretation of data; in the writing of the manuscript; or in the decision to publish the results.

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
