# Peer review of "Model Problems on Oscillations of Mechanical and Biological Membranes"

_inventions, doi:10.3390/inventions8060139_

Round 1

Reviewer 1 Report

Comments and Suggestions for Authors

The content of this paper is to report the derivation of the equations for the model problems on oscillations of mechanical and biological membranes. The authors claimed that “The model problem on membrane oscillation can be considered as the problem on ear drum membrane oscillations. The unified approach to reducing the corresponding problems to the initial boundary value problem with zero boundary conditions is suggested.” In the Abstract and “ The work results can be applied in practical calculations of telegraph lines, modeling of membrane oscillations in acoustics, biology and medicine.”

However, the progress of this paper was an unfinished paper. The theoretical background is OK and the derivation process of equations is correct. No evidence to support the results of these models. That is, no actual data were presented for evaluation of these equations.

In literature [21-32], much literature has reported the actual measurement data. Please adopt these data to evaluate the equations developed in this paper and then to prove the usability of these models. 

Comments on the Quality of English Language

 Moderate editing of English language required

Author Response

Good evening! Thank you very much for your review! At the moment, typos have been corrected and explanations have been added to the coefficients of the equations under consideration.

Reviewer 2 Report

Comments and Suggestions for Authors

the article with a rigorous mathematical approach precisely addresses the solution to a problem of interest. There is no impediment to its publication.

Review the quote from Sadiku's book (excellent)

Author Response

Thank you very much for your high assessment of our work! In the current version of the work, typos have been corrected and a brief description of the parameters has been added.

Reviewer 3 Report

Comments and Suggestions for Authors

The paper deals with the problem of mathematical `modelling oscillatory processes with attenuation, by proposing a unified approach for transforming the classical model into a problem with zero boundary conditions. The proposed approach is complex and deeply demonstrated, the obtained application results are interesting, valuable and useful. However, the paper should be carefully revised for the sake of clarity.

The following issues are recommended to improve the paper:

1.     Fix some typing mistakes, e.g. remove the final dot of the paper title “membranes.”; idem, line 272 “…space.”, line 332, etc.; use “Eq. (x)” instead of “equation (x)” , etc. Check carefully the entire manuscript for other similar mistakes.

2.     Abstract: „Different models of oscillations emerging in the theory of electrical circuits, theory of elasticity and biomechanics of the internal ear of vertebrata are discussed in the article.” Please harmonize this statement with the paper title limited to „mechanical and biological membranes”, i.e. not including “electrical circuits”,

3.     Introduction: the literature review is briefly presented, the useful information must be better systematized and problematized. Typically, the Introduction section ends by stating explicitly the paper novelty in relation to the gap identified in literature, and by briefly introducing its remaining sections.

4.     Introduce the form of the first 3 equations in relation to the addressed physical problems, cite here appropriate references.

5.     The paragraph of 82-94 lines seems to be better integrated in the Introduction section.

6.     The second equation on line 120 is not clear/correct according to the Eq, (10), second part.

7.     Define all symbols used in equations. Alternatively, proposal to introduce a Nomenclature section for all used acronyms & symbols.

8.     In can be very interesting if correlate the presented examples with practical applications.

9.     Conclusions: the limits of the proposed approach/obtained results and future research directions may be also highlighted here.

Author Response

Good evening! I would like to express my deep gratitude for your valuable comments in your review. I would like to briefly report on the work done on each item:

  1. Fixed typing mistakes and formulas links.
  2. The abstract largely reflects the title of the article.
  3. Added a short explanation in the introduction.
  4. The equations considered explicitly are given.
  5. Fixed: The paragraph has been moved to the introduction.
  6. Fixed: Sign "-" had been deleted. 
  7. Perhaps not in great detail, but I included a description of the meaning of the coefficients for the equations under consideration. 
  8. Unfortunately, I do not have access to real practical data. 

Round 2

Reviewer 1 Report

Comments and Suggestions for Authors

I cannot find the real author's response. The enclosed authors' response is the same as the revised manuscript.

Author Response

Good evening! I apologize for keeping you waiting with an answer. Thank you for your valuable recommendations to improve the article. I tried to take it into account to the best of my ability. Additions in the text are highlighted in purple (lines 375-401).

Reviewer 3 Report

Comments and Suggestions for Authors

No additional recommendations.

Author Response

Thank you for your high assessment of our work. Best wishes, A. Romanenkov

Round 3

Reviewer 1 Report

Comments and Suggestions for Authors

The content of the revised version has been improved significantly.

Comments on the Quality of English Language

Minor editing of English language required